# Mayfly: a Neural Data Structure for Graph Stream Summarization

**Yuan Feng**[1,3,†] **, Yukun Cao**[1,3,†]**, Hairu Wang**[1,3]**, Xike Xie**[2,3,*]**, and S. Kevin Zhou**[2,3]

[1]School of Computer Science, University of Science and Technology of China (USTC), China
[2]School of Biomedical Engineering, USTC, China
[3]Data Darkness Lab, MIRACLE Center, Suzhou Institute for Advanced Research, USTC, China
{yfung,ykcho,wanghairu}@mail.ustc.edu.cn, xkxie@ustc.edu.cn,
s.kevin.zhou@gmail.com

## Abstract

A graph is a structure made up of vertices and edges used to represent complex relationships between entities, while a graph stream is a continuous flow of graph updates that convey evolving relationships between entities. The massive volume and high dynamism of graph streams promote research on data structures of graph summarization, which provides a concise and approximate view of graph streams with sub-linear space and linear construction time, enabling real-time graph analytics in various domains, such as social networking, financing, and cybersecurity. In this work, we propose the *Mayfly*, the first neural data structure for summarizing graph streams. The Mayfly replaces handcrafted data structures with better accuracy and adaptivity. To cater to practical applications, Mayfly incorporates two offline training phases, namely larval and metamorphosis phases. During the larval phase, the Mayfly learns basic summarization abilities from automatically and synthetically constituted meta-tasks. In the metamorphosis phase, it rapidly adapts to real graph streams via meta-tasks. With specific configurations of information pathways, the Mayfly enables flexible support for miscellaneous graph queries, including edge, node, and connectivity queries. Extensive empirical studies show that the Mayfly significantly outperforms its handcrafted competitors.

## 1 Introduction

Recently, it shows prominence in using neural networks as alternatives for handcrafted data structures (Cao et al., 2023; Li et al., 2023; Bertsimas & Digalakis, 2021; Rae et al., 2019; Hsu et al., 2019; Kraska et al., 2018), especially in data stream applications, where neural data structures are designed by harnessing the abilities of neural networks, providing two key benefits. First, they exploit deep-level implicit information (*e.g.,* dense vectors), enabling superior summarization capabilities to handcrafted data structures which maintain explicit counts. Second, they facilitate diversified query evaluation over data summarization via data streaming patterns captured in deep networks.

One important but under-explored data stream application for neural data structures is graph streams. A graph stream refers to the continuous sequence of *streaming edges*, each of which is associated with two incident nodes and an edge weight. Graph streams represent the evolutionary process of a dynamic graph and play a vital role in online graphical analytics (McGregor, 2014a; Stanton & Kliot, 2012; Aggarwal et al., 2010; Zou et al., 2022; Kong et al., 2022). For example, in a social network (Mislove et al., 2007; Zhao et al., 2016), graph streams capture the evolving relation between entities, where the edge weight is indicative of the strength of social connections. In network traffic analysis (Guha & McGregor, 2012; D'Alconzo et al., 2019), graph streams highlight the communication between networking devices aiding in monitoring tasks, such as locating security threats (Hong et al., 2017; Gou et al., 2022) or identifying network structures (Ahn et al., 2012; Gou et al., 2022).

Ordinary data streams are with two prominent characteristics, *massive volume* and *high dynamism*, so that challenges arise in meeting the space and time constraints on the stream processing. In particular,

---

† Equal Contribution    *Corresponding Author

it requires data streams to be processed in a *one-pass* fashion with *limited space budgets* (Charikar et al., 2002; Cormode & Muthukrishnan, 2005; Babcock et al., 2002; Broder & Mitzenmacher, 2004). Graph streams post the additional *variety challenge* in sustaining the representation of complex relationships in-between graph entities, putting heavier burdens on contending with the space and time challenges. Existing handcrafted solutions (Cormode & Muthukrishnan, 2005; Zhao et al., 2011; Tang et al., 2016; Gou et al., 2022) summarize graph streams by a small (sub-linear) *summarization structure* within linear construction time and allow for the evaluation of graph queries. Early methods, such as CM-Sketch (Cormode & Muthukrishnan, 2005) and gSketch (Zhao et al., 2011), hash streaming edges to counters of a 2D storage array. However, they overlook the importance of graph connectivity which is essential in addressing advanced graph queries. Afterwards, the TCM (Tang et al., 2016), independently hashes streaming nodes and edges, while preserving the graph connectivity to support diverse graph queries. GSS (Gou et al., 2022) extends the underlying structure of TCM with auxiliary storage to improve the query accuracy. Auxo (Jiang et al., 2023) builds upon GSS and incorporates a prefix embedded tree, to accelerate the process of GSS buffer expansion, particularly in high-budget scenarios. However, the optimization comes at a cost of increased time complexity and its linear memory expansion does not meet the low-budget constraints commonly associated with the majority of existing sketches. In our paper, we emphasize smaller memory budgets (e.g., below 100KB) (Liu & Xie, 2021; 2023). With such budget constraints, existing structures falter. TCM, if constrained by budgets, faces severe hash collisions, causing excessive query errors. Similarly, GSS (Auxo) cannot fetch external storage that exceeds the budgets, thereby resulting in unacceptable query hit rates.

In this paper, we propose the first neural data structure, called the Mayfly, for graph stream summarization, going beyond state-of-the-art handcrafted solutions, such as TCM and GSS. However, challenges arise in determining the appropriate design of network structures and training methods, specifically in meeting the stringent requirements of graph summarization in terms of space and time efficiency, while supporting diverse graph queries, including *edge queries*, *node queries*, *connectivity queries*, *path queries*, and *subgraph queries*.

Starting from the premises, the Mayfly based on *memory-augmented network* (Graves et al., 2014; 2016), is trained by *one-shot meta-learning* (Vinyals et al., 2016; Santoro et al., 2016) based on two types of auto-generated meta-tasks, *larval tasks* and *metamorphosis tasks*, leading to two corresponding training phases, *larval* and *metamorphosis* phases. In the larval phase, the Mayfly learns the fundamental ability to summarize graph streams based on larval tasks generated from synthetic data, focusing on the generalization ability. In the metamorphosis phase, the Mayfly equipped with basic abilities is swiftly adapted to metamorphosis tasks, which are sampled from real graph streams to enhance the specialization ability. The Mayfly adopts a novel method of jointly storing edges/nodes information aligning with the workflow of conventional structures of graph summarization and guaranteeing space efficiency. Moreover, we introduce information pathways coupled with meta-learning guiding offering full coverage of diverse graph query types, liberating the model from full retraining for each type. It simultaneously meets the ubiquitous need to execute multiple query types following one-pass storing in graph streaming scenarios.

**Our contributions include:** We propose the Mayfly, the first neural graph stream summarization structure based on memory-augmented networks with a novel storage paradigm and training strategies tailored for graph stream scenarios. Mayfly takes the graph structure into account and can be rapidly adapted to real graph stream distributions. It introduces a novel concept of altering information pathways in memory networks to facilitate the expansion of multiple graph query types, which hold the potential to facilitate research in other scenarios with diverse query types. Extensive empirical studies show that our proposal significantly outperforms state-of-the-art methods.

## 2 RELATED WORKS

### 2.1 NEURAL DATA STRUCTURE

Mayfly draws inspiration from two recent neural data structures: NBF (Neural Bloom Filter) (Rae et al., 2019) and MS (Meta-Sketch) (Cao et al., 2023). While NBF filters redundant streaming items and MS counts streaming item frequencies, Mayfly introduces significant innovations. Primarily, it is custom-designed for graph stream scenarios, emphasizing separate compressed storage for node and edge weights, to capture graph structural information. In contrast, NBF and MS do not account for

this structural information, making them less apt for such contexts. Additionally, the Mayfly solves the NBF and MS's dependencies on specific datasets by employing randomized IDs for pre-training. A notable addition from Mayfly in neural data structures is the *information pathway*. Modifying these pathways in the memory network facilitates the extension of query types in graph stream contexts.

## 2.2 GRAPH STREAM SUMMARIZATION

As outlined in Section 1, current methods for graph stream summarization primarily hinge on summarization structures. Tracing back, graph stream summarization has evolved from traditional graph summarization, with methods such as *Sparsification* (Li et al., 2022), *Bit compression* (Zhou et al.), *Tensor factorization* (Fernandes et al., 2021), and *GNN-based techniques* (Shabani et al., 2023) having been proposed. But the majority of these methods are designed for static graphs (Liu et al., 2018). If one attempts to customize other dynamic graph methods to support graph streams, restricted query capabilities and significantly reduced efficiency can not be avoided. For example, the classic TimeCrunch (Shah et al., 2015) fails to support basic accumulated weight queries in stream contexts, while the advanced Mosso (Ko et al., 2020) and its variant (Ma et al., 2021) suffer from significantly throughput delays in handling stream tasks, performing at a rate approximately $10^3$ times slower than GSS (Gou et al., 2022). In contrast, the Mayfly's primary focus lies in improving and advancing summarization structures within the context of graph streams.

## 3 CONCEPTS

We formalize graph streams and define atomic queries supported by graph stream summarization to clarify conceptual barriers. A graph stream $\mathcal{S}_G$ (Definition 1) is represented by the directed graph $G(V, E)$, where $V$ denotes the node set and $E$ the edge set. Thus, for an arbitrary edge $e_j \in E$, there exist one or more streaming edges in $\mathcal{S}_G$ with origin node $o(e_j)$ and destination node $d(e_j)$, so that the weight of $e_j$ is equal to the summation of the weights of all corresponding streaming edges.

**Definition 1** (**Graph Streams** (McGregor, 2014b; Gou et al., 2022)). *A graph stream $\mathcal{S}_G$ : $\{x_1, x_2, \ldots\}$ is a time-evolving sequence, where each item $x_i$ represents a **streaming edge** from origin node $o(x_i)$ to destination node $d(x_i)$ with incremental weight $\Delta w_i$ at time $i$.*

The technique stack of graph stream summarization builds upon three atomic query types: *edge, node, and connectivity queries* (Tang et al., 2016; Gou et al., 2022). Among these, the *edge query*, which retrieves the weight of a specified edge (Definition 2), is of paramount importance, since it serves as the basis for evaluating other query types. Mayfly is primarily trained on meta-tasks related to *edge queries*. Once trained, the Mayfly can be swiftly transited to the processing of *node queries* and *connectivity queries* (Definition 3 and 4 ). More complex queries, such as *path and subgraph queries*, can be disassembled to a series of atomic queries. For simplicity, we investigate the Mayfly training guided by edge queries in Section 4 and study the extensions to other query types in Section 6.

**Definition 2** (**Edge Queries**). *An edge query, denoted as $Q_w(e_j)$, aims to retrieve the cumulative weight of a given edge $e_j$ on graph $G(V, E)$ rendered by graph stream $\mathcal{S}_G$: $Q_w(e_j) = \sum_{i|x_i \in \mathcal{S}, o(x_i)=o(e_j), d(x_i)=d(e_j)} \Delta w_i$.*

**Definition 3** (**Node Queries**). *Given a node $v_m$ of graph $G(V, E)$ rendered by graph stream $\mathcal{S}_G$, a node query aims to retrieve either the sum of the cumulative weights of $v_m$'s outgoing edges or its incoming edges: $Q_{n_{out}}(v_m) = \sum_{i|x_i \in \mathcal{S}, o(x_i)=v_m} \Delta w_i$ or $Q_{n_{in}}(v_m) = \sum_{i|x_i \in \mathcal{S}, d(x_i)=v_m} \Delta w_i$.*

**Definition 4** (**Connectivity Queries**). *Given two nodes $v_m$ and $v_{m'}$ in a graph $G(V, E)$ represented by graph stream $\mathcal{S}_G$, a connectivity query determines if there's a directed edge $e_j$ from node $v_m$ to node $v_{m'}$. Formally, $Q_c(v_m, v_{m'}) = $ True if $e_j \in E$ and False otherwise.*

## 4 METHODOLOGY

### 4.1 OVERVIEW

Overall, the Mayfly framework utilizes a storage matrix $\mathcal{M}$ as the foundation for summarizing graph streams by writing sequentially arriving streaming edges into it in one pass. Built upon the

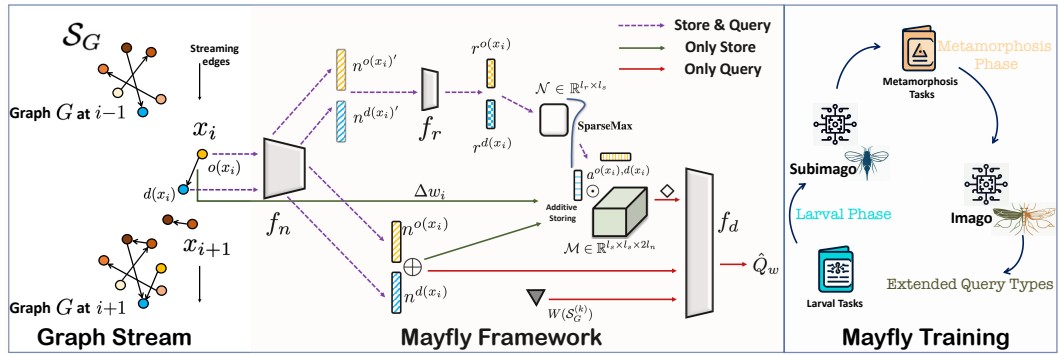

Figure 1: Mayfly Overview

storage matrix, the framework consists of three functional modules, namely *Representation* ($\mathcal{R}$), *Addressing* ($\mathcal{A}$), and *Decoding* ($\mathcal{D}$), as shown in Figure 1. Similar to traditional summarization structures, we define two types of operations for the Mayfly in association with the three modules, which are a unified *Store* operation and a customizable *Query* operation. Specifically, the *Store* operation first inputs each incoming streaming edge into $\mathcal{R}$ for representation learning, considering the joint information of the two incident nodes of the incoming edge. Then, based on the address obtained from $\mathcal{A}$, the representation vector and corresponding weights are written into $\mathcal{M}$. The *Query* operation can be instantiated for different types of graph stream queries via the specific configuration of information pathways. Remarkably, different *Query* operation instances generally follow a similar process, where they utilize the address from $\mathcal{A}$ to read information vectors from $\mathcal{M}$ and decode the retrieved information through $\mathcal{D}$ to get the query result.

The Mayfly adopts the idea of meta-learning, which involves learning the ability to solve a class of domain tasks from auto-generated meta-tasks, rather than memorizing a specific task. To satisfy the one-pass processing for graph streams, we employ a one-shot meta-training strategy (Rae et al., 2019; Vinyals et al., 2016), which enables efficient meta-optimization within a single base learning episode (Hospedales et al., 2021). Moreover, the Mayfly benefits from the paradigm of *"pre-training and fine-tuning"*, consisting of two training phases, the *larval phase* and the *metamorphosis phase*, to gracefully handle the complex real-world graph stream applications. In the larval phase, the Mayfly learns the fundamental techniques of summarizing graph streams by undergoing the training with *larval tasks*, to reach the *subimagio state*. In the metamorphosis phase, the subimago form of the Mayfly quickly transforms to the *imago state*, by rapid adapting to the *metamorphosis tasks*, to gain the ability on the real graph streams.

## 4.2 MODULES

**Representation ($\mathcal{R}$).** The main function of module $\mathcal{R}$ is to learn representations for an incoming streaming edge $x_i$, while edges are represented in the same way during the *Query* operation. We use a set of representation vectors for node $o(x_i)$ and $d(x_i)$, denoted as $\{n^{o(x_i)}, n^{d(x_i)}\}$, to represent streaming edge $x_i$. The information of a streaming edge is implicitly encoded in these representation vectors (i.e., "*representing*" function), which also serve as the basis for deriving the certain read/write addresses (i.e., "*addressing*" function). Specifically, $\mathcal{R}$ contains a network component, the representation network $f_n$. For a streaming edge $x_i$, its out/in nodes $o(x_i), d(x_i) \in V$ is numerically encoded and then input to $f_n$ to get $\{n^{o(x_i)}, n^{d(x_i)}\} \in \mathbb{R}^{l_n}$. The out/in bidirectional representation adequately represents the stacked edges while avoiding confusion in both the outgoing and incoming directions.

$$\mathcal{R}[x_i] \to \mathcal{R}[\{o(x_i), d(x_i)\}] \to \{f_n(o(x_i)), f_n(d(x_i))\} \to \{n^{o(x_i)}, n^{d(x_i)}\} \quad (1)$$

We adopt a unified encoding for nodes of a graph stream, where for each node a unique numerical ID (*e.g.,* $1, 2, \ldots$) is assigned and then converted to binary encoding. The unified encoding strategy allows the Mayfly to be independent of node information in a graph stream, conducting pre-training independent of real dataset. By establishing the same encoding mapping for various streams, the Mayfly can be trained once and deployed for multiple applications without being retrained.

**Addressing ($\mathcal{A}$).** The module $\mathcal{A}$ aims to derive the read/write addresses on storage matrix $\mathcal{M}$ for each streaming edge $x_i$. To reconcile the conflict between the "*representing*" and "*addressing*" functions of the representation vectors and increase the diversity of network pathways, we extract the intermediate

representations (i.e., output results of a specific hidden layer in $f_n$) of $\{f_n(o(x_i)), f_n(d(x_i))\}$, denoted as $\{n^{o(x_i)'}, n^{d(x_i)'}\}$, and input them into a network component $f_r$ to obtain the refined vector set $\{f_r(n^{o(x_i)'}), f_r(n^{d(x_i)'})\} \to \{r^{o(x_i)}, r^{d(x_i)}\} \in \mathbb{R}^{l_r}$ as the basis of addressing,

$$\mathcal{A}[\{n^{o(x_i)'}, n^{d(x_i)'}\}] \to \textbf{SparseMax}(\{\beta \cdot (f_r(n^{o(x_i)'}))^T \mathcal{N}, \beta \cdot (f_r(n^{d(x_i)'}))^T \mathcal{N}\}) \to \{a^{o(x_i)}, a^{d(x_i)}\} \quad (2)$$

Here, $\mathcal{N} \in \mathbb{R}^{l_r \times l_s}$ is an addressing matrix with learnable parameters, which can be viewed as a differentiable simulation of the hashing process used in traditional summarization structures (Rae et al., 2019). Then, the matrix multiplication of $\mathcal{N}$ and the transpose of $\{r^{o(x_i)}, r^{d(x_i)}\}$ results in the addresses $a^{o(x_i)}$ and $a^{d(x_i)}$. The two addresses, corresponding to out/in direction, jointly determine the storage positions of the representation vectors (i.e., $\{n^{o(x_i)}, n^{d(x_i)}\}$) of a streaming edge within $l_s{}^2$ slots in the memory matrix $\mathcal{M} \in \mathbb{R}^{l_s \times l_s \times 2l_n}$. Specifically, we treat $a^{o(x_i)}$ as the *row address* and $a^{d(x_i)}$ as the *column address*, and determine the specific storage positions $a^{o(x_i),d(x_i)} \in \mathbb{R}^{l_s \times l_s}$ of $\{n^{o(x_i)}, n^{d(x_i)}\}$ in $\mathcal{M}$. In combination with the out/in bidirectional representation vectors of length $2l_n$ in one slot for different edges, $\mathcal{M}$ provides a joint compression storage for nodes and edges. This efficient and interpretable novel storage paradigm not only provides sufficient information for decoding, but also lays the foundation for the extension of multiple graph query types as shown in Section 6. Moreover, we utilize a *Sparse SoftMax* (Laha et al., 2018; Martins & Astudillo, 2016) for the normalization of addresses to reduce the noise in information stacking [1]. The $\beta$ is a learnable scale value that controls the sparsity of addresses, which is verified in Appendix A.6.

**Decoding** ($\mathcal{D}$). Given a query edge $e_j$, the module $\mathcal{D}$ aims to decode the stacked information of $e_j$ read from $\mathcal{M}$ based on address $a^{o(e_j),d(e_j)}$ to obtain the corresponding query result. The module has one network component $f_d$,

$$\mathcal{D}[a^{o(e_j),d(e_j)} \diamond \mathcal{M}, n^{o(e_j)} \oplus n^{d(e_j)}, W(\mathcal{S}_G)] \to f_d(a^{o(e_j),d(e_j)} \diamond \mathcal{M}, n^{o(e_j)} \oplus n^{d(e_j)}, W(\mathcal{S}_G)) \to \hat{Q}_w(e_j) \quad (3)$$

Here, the representation vectors $n^{o(e_j)}$ and $n^{d(e_j)}$ for $e_j$, along with the total sum of all incremental weights of the graph stream $W(\mathcal{S}_G)$, are fed into $f_d$ as auxiliary decoding information, where symbol $\oplus$ denotes the vector concatenation operation. The $a^{o(e_j),d(e_j)} \diamond \mathcal{M}$ represents the operation for reading $e_j$'s information from matrix $\mathcal{M}$. In this paper, the implementation of $\diamond$ adheres to the classical *content-based read mechanism* (Wu et al., 2018; Rae et al., 2019; Graves et al., 2016; 2014), and incorporates an auxiliary *MinGain* term:

$$a^{o(e_j),d(e_j)} \diamond \mathcal{M} \to \underbrace{a^{o(e_j),d(e_j)} \otimes \mathcal{M}}_{\textit{Content-Based Read}} + \underbrace{\textbf{min}(|a^{o(e_j),d(e_j)} \otimes \mathcal{M}|_{n^{o(e_j)} \oplus n^{d(e_j)}})}_{\textit{MinGain Term}} \quad (4)$$

Here, the $\otimes$ represents the summation of all slots in $\mathcal{M}$ weighted by address $a^{o(e_j),d(e_j)}$. The *MinGain* term[1] represents the minimum value in $a^{o(e_j),d(e_j)} \otimes \mathcal{M}$ after normalization based on $n^{o(e_j)} \oplus n^{d(e_j)}$, aiming to extract the crucial low numerical value bits for decoding but are easily overshadowed by stacked noise from other large value bits. The detailed formalization is outlined in Appendix A.1.

### 4.3 OPERATIONS

• Operation **Store** is executed by feeding an incoming streaming edge $x_i$ into $\mathcal{R}$ and $\mathcal{A}$ to obtain representation vectors $\{n^{o(x_i)}, n^{d(x_i)}\}$ and address $a^{o(x_i),d(x_i)}$. Then, concatenation vector is additively stored to $\mathcal{M}$ after being multiplied by incremental weight $\Delta w_i$,

$$\mathcal{M} = \mathcal{M} + a^{o(x_i),d(x_i)} \odot (n^{o(x_i)} \oplus n^{d(x_i)}) \cdot \Delta w_i \quad (5)$$

Here, $\odot$ represents element-wise matrix multiplication. Additive storing (Rae et al., 2019) is an efficient and commonly used method in memory networks. Note that the out/in bidirectional storage of origin node information $n^{o(x_i)}$ and destination node information $n^{d(x_i)}$ are independent in one memory slot(i.e., first $l_n$ bits are for $n^{o(x_i)}$ and last $l_n$ bits are for $n^{d(x_i)}$).

• Operation **Query** estimates the cumulative weight of a given query edge $e_j$ after storing a graph stream $\mathcal{S}_G$. Firstly, representation vectors $\{n^{o(e_j)}, n^{d(e_j)}\}$ and the address $a^{o(e_j),d(e_j)}$ are obtained, following a similar process of the Store operation. Then, $\{a^{o(e_j),d(e_j)} \diamond \mathcal{M}\}$, $n^{o(e_j)} \oplus n^{d(e_j)}$ and the summation of all weights $W(\mathcal{S}_G)$ are jointly input into $\mathcal{D}$ to obtain the estimated weight $\hat{Q}_w(e_j)$. For detailed description about operations, please refer to Appendix A.1.

---

[1]Relevant ablation experiments can be found in Appendix A.7

## 4.4 TRAINING

The Mayfly employs a meta-learning training algorithm to endow effective parameters in all learnable network modules, $f_n$, $f_a$, $\mathcal{A}$, and $f_d$. Adhering to the standard setup of meta-learning, the fundamental unit of the Mayfly training is the meta-task, which guides the latent optimization direction. In our paper, the Mayfly consists of two offline training phases in correspondence to two types of meta-tasks, larval tasks and metamorphosis tasks. Both phases share the same training algorithm. And the formal algorithms for training and generating meta tasks are discribed in Appendix A.2.

**Training Algorithm.** During training process, the Mayfly iterates over the set of larval tasks or metamorphosis tasks. We can view the training process on a single task as simulating the storing and querying of a graph stream instance while computing the error for optimizing the learnable modules. Therefore, a (larval or metamorphosis) task $t_k$ consists of two parts, a support set $s_k$ and a query set $q_k$. The support set $s_k$:$\{x_1^{(k)}, x_2^{(k)}, \dots\}$ represents a graph stream instance $\mathcal{S}_G^{(k)}$ with streaming edges $\{x_i^{(k)}\}$, while the query set $q_k$:$\{e_1^{(k)} : Q_w(e_1^{(k)}), \dots\}$ represents the edges $\{e_j^{(k)}\}$ to be queried along with their query results $\{Q_w(e_j^{(k)})\}$. We use the balanced (relative) mean squared error as the loss function $\mathcal{L}$ with learned parameters $\lambda_1$ and $\lambda_2$ (Kendall et al., 2018),

$$\mathcal{L} \rightarrow (Q_w(e_j) - \hat{Q}_w(e_j))^2 / 2\lambda_1^2 + |Q_w(e_j) - \hat{Q}_w(e_j)| / (2\lambda_2^2 Q_w(e_j)) + \log \lambda_1 \lambda_2 \qquad (6)$$

**Larval Task Generation.** In the larval phase, the target of larval tasks is to train the Mayfly with basic abilities to summarize graph streams while maintaining generality to different graph streams. In practical applications, the distribution of edge weights in a graph often follows skewed distributions, especially the *Zipf* distributions (Kali, 2003; Chen & Wang, 2010; Aiello et al., 2002). So, we constitute larval tasks by making the edge weights follow a wide range of Zipf distributions with varied parameter $\alpha$. The family of Zipf distributions constitute a distribution pool $\mathcal{P}$. Note that the Mayfly does not rely on the clumsy memorization of larval tasks. The basic summarization capabilities learned in this phase apply for diverse graph streams beyond the larval tasks.

A larval task $t_k$ is generated through three steps, which essentially synthesize a graph stream $\mathcal{S}_G^{(k)}$. First, we synthesize streaming edges $\{x_i^{(k)}\}$ with the stream length $|\mathcal{S}_G^{(k)}| \in [1, \gamma]$, where the two nodes of a streaming edge are randomly sampled and encoded from a unified numerical ID space. Second, a distribution instance $p^{(k)}$ is sampled from the distribution pool $\mathcal{P}$. For each $x_i^{(k)}$, a weight is assigned, which is obtained by the product of a total weight sum $W(\mathcal{S}_G^{(k)})$ and a sample from the distribution $p^{(k)}$. Finally, the synthetic graph stream $\mathcal{S}_G^{(k)}$ serves as the support set $s_k$, and the query result $Q_w(e_j^{(k)})$ for all $e_j^{(k)}$s constitutes the query set $q_k$.

**Metamorphosis Task Generation.** In the metamorphosis phase, the metamorphosis task aims to capture and reflect typically skewed patterns found in real graph streams. If the empirical weight distribution is available (e.g., from historical records or sampled episodes of real graph streams), it can be used to generate the metamorphosis tasks. To generate metamorphosis tasks, we extract consecutive portions of different lengths from the real stream. In addition, we intentionally blur the relationship between edges and their weights in these extracted portions. This deliberate blurring allows the metamorphosis tasks to effectively represent the concept drift characteristics of the stream (see extra experiments in Appendix A.9).

## 5 EXPERIMENTS

### 5.1 SETUP

**Datasets.** We use four commonly used public graph stream datasets, comprising two *medium-sized* datasets (Lkml, Enron) and two *large-scale* datasets (Coauthor, Twitter). The Lkml (Gou et al., 2022; Xu & Zhou, 2018) and Enron Shin et al. (2020); Lee et al. (2020) pertain to communication/social networks. Specifically, Lkml contains 1,096,440 communication records exchanged among 63,399 Linux kernel mail network nodes, and Enron captures 475,097 email exchanges across 17,733 addresses. Coauthor (Newman, 2001) represents the co-authorship social network among researchers, embodying 2,668,965 streaming edges. The original Twitter dataset is static with no duplication, encompassing 52,579,682 nodes and 1,963,263,821 edges. We follow the setting of (Gou et al., 2022),

Table 1: Results in Lkml & Enron Dataset

| Model | Length | Lkml Dataset | | | | Enron Dataset | | | |
|---|---|---|---|---|---|---|---|---|---|
| | | 20K | 80K | 200K | 800K | 20K | 40K | 200K | 400K |
| TCM | ARE | 1.26 ±0.09 | 6.11 ±0.22 | 15.89 ±0.48 | 66.30 ±0.13 | 1.77 ±0.02 | 3.71 ±0.03 | 16.01 ±0.02 | 25.07 ±0.01 |
| (B=64KB) | AAE | 1.93 ±0.14 | 9.84 ±0.25 | 26.11 ±0.88 | 110.56 ±0.38 | 2.08 ±0.02 | 4.56 ±0.03 | 24.13 ±0.03 | 45.06 ±0.01 |
| Subimago | ARE | 6.24 ±1.97 | 11.21 ±4.07 | 18.12 ±5.83 | 42.96 ±1.18 | 3.80 ±0.03 | 4.13 ±0.03 | 7.51 ±0.05 | 10.00 ±0.01 |
| (B=64KB) | AAE | 9.05 ±3.20 | 18.33 ±7.19 | 31.16 ±11.14 | 77.07 ±2.35 | 4.30 ±0.03 | 4.91 ±0.03 | 11.13 ±0.04 | 17.73 ±0.01 |
| Imago | ARE | **0.94 ±0.05** | **1.33 ±0.06** | **2.07 ±0.07** | **5.86 ±0.01** | **1.00 ±0.01** | **1.10 ±0.01** | **2.37 ±0.01** | **3.42 ±0.01** |
| (B=64KB) | AAE | **1.57 ±0.08** | **2.62 ±0.12** | **3.92 ±0.12** | **9.62 ±0.03** | **1.18 ±0.01** | **1.43 ±0.01** | **3.72 ±0.00** | **6.14 ±0.01** |
| TCM | ARE | 0.52 ±0.05 | 2.77 ±0.10 | 7.34 ±0.19 | 31.34 ±0.05 | 0.66 ±0.01 | 1.50 ±0.01 | 6.90 ±0.01 | 10.82 ±0.01 |
| (B=128KB) | AAE | **0.81 ±0.08** | 4.55 ±0.11 | 12.34 ±0.29 | 53.44 ±0.16 | 0.79 ±0.01 | 1.87 ±0.02 | 10.57 ±0.01 | 19.82 ±0.01 |
| Subimago | ARE | 2.43 ±0.37 | 4.30 ±0.60 | 6.46 ±0.86 | 16.89 ±0.20 | 2.58 ±0.02 | 3.22 ±0.02 | 4.93 ±0.02 | 6.37 ±0.01 |
| (B=128KB) | AAE | 3.36 ±0.62 | 6.78 ±1.13 | 10.76 ±1.68 | 29.03 ±0.36 | 2.89 ±0.03 | 3.80 ±0.03 | 7.34 ±0.02 | 11.43 ±0.01 |
| Imago | ARE | **0.50 ±0.02** | **0.77 ±0.03** | **1.00 ±0.03** | **2.51 ±0.00** | **0.31 ±0.01** | **0.60 ±0.01** | **1.67 ±0.01** | **2.42 ±0.01** |
| (B=128KB) | AAE | 0.96 ±0.03 | **1.86 ±0.08** | **2.63 ±0.08** | **5.17 ±0.02** | **0.50 ±0.01** | **0.93 ±0.01** | **2.88 ±0.01** | **4.70 ±0.01** |

which assigns weights to all edges using Zipf distributions, resulting in 10,929,205,002 streaming edges. These datasets exhibit varying degrees of deviation from the ideal Zipf distributions. Please refer to the Appendix A.4 for the visualization of dataset statistics. In addition, each node in the four datasets is assigned a numerical ID in binary encoding.

**Baselines.** We have chosen TCM and GSS as the competitors, which are SOTA under small budgets. Two commonly accepted metrics, average absolute error (AAE) and average relative error (ARE), are adopted for evaluation. The code for Mayfly has been included in the supplementary materials. GSS and TCM, utilize the open-source code provided alongside their papers.

**Parameters.** We have implemented $f_n$ and $f_r$ in MLP with 5-layer and 2-layer of size 32, followed by the batch normalization. The intermediate representation of the third layer of $f_n$ is fed into $f_r$ as inputs. The $f_d$ is implemented in a 3-layer MLP with residual connections, where each layer is of size 64. $Relu$ function is chosen for layer connections. The space budget ($B$) of the Mayfly is allocated to matrix $\mathcal{M}$ following the common setting of neural data structures (Rae et al., 2019). Further parameter details, including module sizes and settings of $\mathcal{M}$, can be found in Appendix A.5.

**Larval Phase.** We set $\gamma = 60,000$ and use the Zipf distributions with $\alpha$ ranging from 0.3 to 0.8 to build the distribution pool $\mathcal{P}$. The total weight sum is ranging from 5 to 50 times of the edges in graph. The number of training steps is 500,000 and the learning rate is 0.0005.

**Metamorphosis Phase.** We split each dataset into $D_{train}$ and $D_{test}$, using a 2:8 based on timestamps. By doing so, there must be a large population of streaming edges in $D_{test}$ that never appear in $D_{train}$, for examining the generality of the Mayfly. $D_{train}$ is used for metamorphosis task generation, and $D_{test}$ is used for testing. The number of training steps is only 10% of that in the larval phase.

## 5.2 RESULTS ON REAL DATASETS

We extract a set of sub-streams with different lengths from $D_{test}$ of two medium-sized real dataset, Lkml and Enron, for testing how the data summarization technique scales. Table 1 shows the performance on Lkml and Enron, where imago dominates TCM in all testing cases. For example, the AAE of TCM is about 45 (Enron, B=64KB, length=400K), and the AAE of imago is only 6, which is 6.5 times lower. Also, it shows that all methods degrade w.r.t. the stream length, while the performances of imago and subimago are quite stable, compared to the dramatic degrading of TCM. For example, when increasing the stream length from 20K to 800K (Lkml, B=64KB), the AAE value of TCM increases about 52 times, while the AAE of imago only increases about 5 times, which is an order of magnitude lower. An interesting point is that subimago outperforms TCM, when the stream length is higher than 200K, demonstrating the strong zero-knowledge generality of Mayfly.

We evaluate the Mayfly's performance on two large-scale datasets. For million-scale Coauthor, we deploy a 256KB-sized Mayfly. For billion-scale Twitter dataset, we adopt a pragmatic approach to circumvent the extensive training time of a higher-budgeted Mayfly. Specifically, we employ a simple hash function, which evenly maps the streaming edges over a cluster of 128KB-sized Mayfly instances. The space budget is 0.5GB, a reasonable but relatively small allocation, as in previous settings (Tang et al., 2016; Gou et al., 2022). Table 2 shows Mayfly consistently outperforms TCM.

We proceed to compare the performance of our method with another SOTA, GSS. We find that GSS incurs considerable missing hits when the budget is tight, so that the result is often not found for low-weight querying edges. For example, in Figure 2, the hit rate of GSS is less than 20% in all cases, while the hit rates of TCM and imago are 100%. To force the comparison happen, we consider

Table 2: Results in Coauthor & Twitter Dataset

| Model | TCM | | Subimago | | Imago | |
|---|---|---|---|---|---|---|
| Dataset | ARE | AAE | ARE | AAE | ARE | AAE |
| Coauthor(B=256KB,Length=2,000K) | 13.41 ±0.01 | 61.70 ±0.01 | 4.51 ±0.01 | 19.32 ±0.01 | **1.54 ±0.01** | **8.81 ±0.01** |
| Twitter(B=0.5GB,Length=10,000,000K) | 63.02 ±0.12 | 245.25 ±0.51 | 16.30 ±0.08 | 63.53 ±0.37 | **2.18 ±0.01** | **8.50 ±0.05** |

Table 3: Results in Synthetic Datasets

| | B | 64KB | | | 128KB | | |
|---|---|---|---|---|---|---|---|
| | Zipf | 0.2 | 0.6 | 1 | 0.2 | 0.6 | 1 |
| TCM | ARE | 15.45 ±0.02 | 18.12 ±0.02 | 20.70 ±0.06 | 6.71 ±0.01 | 7.61 ±0.01 | 8.36 ±0.01 |
| | AAE | 33.13 ±0.09 | 77.50 ±0.08 | 41.21 ±0.08 | 38.58 ±0.03 | 32.54 ±0.03 | 16.66 ±0.02 |
| SubImago | ARE | 1.23 ±0.01 | 2.05 ±0.01 | 5.31 ±0.24 | 1.01 ±0.01 | 1.72 ±0.01 | 4.45 ±0.03 |
| | AAE | 6.92 ±0.01 | 8.48 ±0.02 | 10.58 ±0.44 | 5.61 ±0.01 | 7.14 ±0.01 | 8.80 ±0.08 |
| ImagoLkml | ARE | **0.44 ±0.01** | **0.97 ±0.03** | 4.82 ±1.47 | 0.39 ±0.01 | **0.38 ±0.01** | 2.28 ±0.56 |
| | AAE | **2.54 ±0.04** | **4.68 ±0.13** | 10.41 ±2.92 | 2.40 ±0.01 | **2.83 ±0.03** | 5.55 ±1.13 |
| ImagoEnron | ARE | 0.48 ±0.01 | 1.06 ±0.01 | **3.37 ±0.01** | **0.17 ±0.01** | 0.60 ±0.01 | **2.21 ±0.05** |
| | AAE | 2.66 ±0.01 | 4.82 ±0.02 | **7.17 ±0.07** | **1.05 ±0.01** | 3.21 ±0.01 | **4.92 ±0.09** |

the heavy edge query (Tang et al., 2016), a variant of the edge query which considers high-weight edges. We compute the top 5% and top 10% heavy edges from the original data as the ground truth. For each of the heavy edges, we retrieve its weight from imago, TCM, and GSS, and compare them with the ground truth. The results on Lkml and Enron are shown in Figure 2 (a) and (b), respectively. The result shows that the performance of imago dominates those of TCM and GSS. For example, the AAE value of imago is about 14.5, while the AAE values of GSS and TCM are 23.9 and 55.6, respectively (Enron, B=64KB, length= 400K).

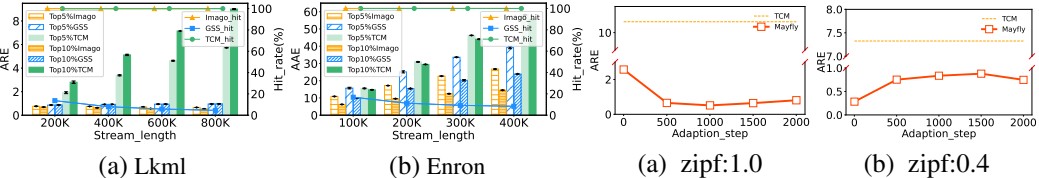

(a) Lkml    (b) Enron    (a) zipf:1.0    (b) zipf:0.4

Figure 2: Heavy Edge Queries and Hit Rates    Figure 3: Concept Drifts

## 5.3 RESULTS ON SYNTHETIC DATASET

To investigate Mayfly's performance in potential data stream concept drift (i.e., distributional shift) scenarios, we evaluate subimago on synthetic graph streams of length 500K, where the accumulative weight of an edge follows Zipf distributions with $\alpha \in \{0.2, 0.6, 1.0\}$. Note that 0.2 and 1.0 are not covered by the distribution pool $\mathcal{P}$ in the larval phase, for testing the generalization. We also examine the performance of imagos, which have been adapted to Lkml and Enron, on synthetic datasets. Table 3 demonstrates the result on synthetic graph streams, where subimago outperforms TCM in all testing cases, highlighting its robust zero-knowledge generalization capabilities. For example, when Zipf parameter $\alpha$ is 0.2 with 64KB budget, the ARE of subimago is 1.23, whereas the ARE of TCM is 15.45. Furthermore, the performance of imago remains impressive, indicating that Mayfly retains its robust generalization capabilities, even after adapting to specific distributions.

To delve deeper into concept drifts, we show the real-time performance of imago on several different distributions, when adapting to a specific Zipf distribution $\alpha$=1.0. The Figure 3 along with the Figure 12 in Appendix A.12 shows a significant improvement with the adapted Zipf distribution $\alpha$=1.0. Meanwhile there is a slight fluctuation with other distributions $\alpha \in \{0.4, 0.6, 0.8\}$, imago still exhibits superior performance. For more analysis of concept drifts, please refer to Appendix A.9.

## 6 EXTENSION

### 6.1 EXTENSION TO ATOMIC QUERIES BY CONSTRUCTING NEW INFORMATION PATHWAY

After the metamorphosis phase, the imago form of the Mayfly has acquired the capability in summarizing graph streams and handling *edge queries*, laying down a foundation for other types of queries. We study on extending the Mayfly to two other new types of atomic queries on graph streams, *node queries* and *connectivity queries* by constructing new information pathways coupled with adding new decoding modules, rather than starting the entire training process from scratch.

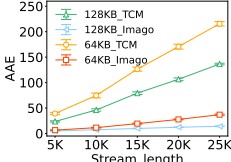 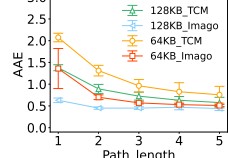 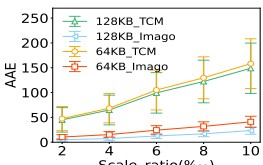

Figure 4: Connectivity Query    Figure 5: Node Query    Figure 6: Path Query    Figure 7: Subgraph Query

Firstly, we craft new information pathways for the new atomic queries, which determine how the information in $\mathcal{M}$ is utilized. Subsequently, we introduce new Decoding modules trained on relevant meta-tasks for new atomic queries. This approach enables Mayfly to adapt to new atomic query types by solely updating the new Decoding modules while keeping other module parameters unchanged.

We elaborate this extension strategy using *node queries* which have significantly different information pathways from the *edge queries*. Firstly, for a given query node $v_m$, we derive vector $n^{v_m}$ and address $a^{v_m}$ using the representation and addressing modules. As described in Section 4.3, the source and destination node information of streaming edges are stored separately within a single slot of $\mathcal{M}$. Specifically, the source node information occupies the first $l_n$ positions of the slot, while the destination node information occupies the remaining $l_n$ positions. Therefore, $Q_{n_{out}}(v_m)$ is relevant to the slice $\mathcal{M}^o \in \mathbb{R}^{l_s \times l_s \times l_n}$ constituted by the first $l_n$ of $2l_n$ positions for all slots in $\mathcal{M}$, and $Q_{n_{in}}(v_m)$ is relevant to the slice constituted by the rest $l_n$ positions, $\mathcal{M}^d = \mathcal{M} - \mathcal{M}^o$. Next, the retrieved information $a^{v_m} \diamond \mathcal{M}^o$ (or $a^{v_m} \diamond \mathcal{M}^d$), along with the auxiliary information $n^{v_m}$ and $W(\mathcal{S}_G)$, are jointly input into the decoding module to obtain the estimated $\hat{Q}_{n_{out}}$ (or $\hat{Q}_{n_{in}}$):

$$\mathcal{D}[a^{v_m} \diamond \mathcal{M}^{o/d}, n^{v_m}, W(\mathcal{S}_G)] \to f_d(a^{v_m} \diamond \mathcal{M}^{o/d}, n^{v_m}, W(\mathcal{S}_G)) \to \hat{Q}_{n_{out/in}} \quad (7)$$

Figure 5 shows the performance of the imago on $Q_{n_{out}}$ on Lkml, which significantly outperforms that of TCM, especially when the length of the graph stream increases.

The operation of connectivity queries is similar to that of edge queries with the sole distinction being that the output of the decoding module takes the form of binary classification labels indicating the connectivity. In addition, we examine the query accuracy for high-weight edges, which are more significant in the graph (Tang et al., 2016). We use edges with top 50%, 30%, and 10% weights as positive samples in test tasks. Figure 4 shows the accuracy of the extended imago form of the Mayfly on the test tasks, based on $D_{test}$ of the Lkml dataset. It demonstrates that imago exhibits a more stable performance, especially for high-weight connectivity queries over varied stream lengths. Conversely, TCM exhibits insensitivity to edge weights and incurs a notable performance drop.

## 6.2 EXTENSION TO MISCELLANEOUS QUERIES BY INVOKING ATOMIC QUERIES

Consistent with TCM and GSS, we could amalgamate multiple invocations of the above atomic queries to cater to a diverse range of graph queries. We construct and evaluate the path queries and subgraph queries (Tang et al., 2016) using *edge queries* and *connectivity queries* as the main building blocks. The graph path queries aim to find the maximum flow along a given path which is a sequence of directed edges. The subgraph queries retrieve the aggregated weight for the edges within the subgraph. For detailed information about these query semantics, please refer to Appendix A.10. Figures 6 and 7 demonstrate the results of two queries (Lkml, length = 20K). It shows that imago outperforms TCM on path queries in all cases. Also, imago exhibits superior performance to TCM in subgraph queries, showing much better stability as the size of the subgraph increases.

## 7 CONCLUSION

In this paper, we propose the first neural data structure for graph stream summarization, called the Mayfly. The Mayfly is based on memory-augmented networks and meta-learning, allowing it to be pretrained through automatic and synthetic meta-tasks, and rapidly adapt to real data streams. With specific configurations of information pathways, the Mayfly enables flexible support for a broad range of graph queries, including edge, node, path, and subgraph queries. We conduct extensive experiments and detailed analysis to demonstrate the superiority of the Mayfly compared to its handcrafted competitors and gain insights into our proposals.

ACKNOWLEDGMENTS

This work is supported by NSFC (No.61772492, 62072428).

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

## A APPENDIX

### A.1 DETAILS OF MAYFLY OPERATIONS

Algorithm1 describes the details of Mayfly's operations on a (stream) edge $x_i$ or $e_j$, including dimensions conversion operations and broadcast operations.

---

**Algorithm 1:** Details of Mayfly Operations

---

**Operation** `Store`$(o(x_i), d(x_i), \Delta w_i, \mathcal{M})$:

  $n^{o(x_i)}, n^{o(x_i)'}, n^{d(x_i)}, n^{d(x_i)'} \leftarrow$ `Representation`$(\mathcal{R})\,(o(x_i), d(x_i))$

  $a^{o(x_i)}, a^{d(x_i)} \leftarrow$ `Addressing`$(\mathcal{A})\,(n^{o(x_i)'}, n^{d(x_i)'})$

  $a^{o(x_i),d(x_i)} \leftarrow a^{o(x_i)}(a^{d(x_i)})^T$

  $n^{x_i} \leftarrow n^{o(x_i)} \oplus n^{d(x_i)}$

  $n^{x_i} \leftarrow ChangeShape(n^{x_i}, \mathbb{R}^{2l_n}, \mathbb{R}^{l_s \times l_s \times 2l_n})$

  $a^{o(x_i),d(x_i)} \leftarrow ChangeShape(a^{o(x_i),d(x_i)}, \mathbb{R}^{l_s \times l_s}, \mathbb{R}^{l_s \times l_s \times 2l_n})$

  $\mathcal{M} = \mathcal{M} + a^{o(x_i),d(x_i)} \odot n^{x_i} \cdot \Delta w_i$

**Operation** `Query`$(o(e_j), d(e_j), \mathcal{M}, W(S_G))$:

  $n^{o(e_j)}, n^{o(e_j)'}, n^{d(e_j)}, n^{d(e_j)'} \leftarrow$ `Representation`$(\mathcal{R})\,(o(e_j), d(e_j))$

  $a^{o(e_j)}, a^{d(e_j)} \leftarrow$ `Addressing`$(\mathcal{A})\,(n^{o(e_j)'}, n^{d(e_j)'})$

  $a^{o(e_j),d(e_j)} \leftarrow a^{o(e_j)}(a^{d(e_j)})^T$

  $\hat{Q}_w(e_j) \leftarrow$ `Decoding`$(\mathcal{D})\,(\{a^{o(e_j),d(e_j)} \diamond \mathcal{M}\}, n^{o(e_j)} \oplus n^{d(e_j)}, W(S_G));$

  **return** $\hat{Q}_w(e_j);$

**Module** `Representation`$(\mathcal{R})\,(o(x_i), d(x_i))$:

  $n^{o(x_i)}, n^{o(x_i)'} \leftarrow f_n(o(x_i))$

  $n^{d(x_i)}, n^{d(x_i)'} \leftarrow f_n(d(x_i))$

  **return** $n^{o(x_i)}, n^{o(x_i)'}, n^{d(x_i)}, n^{d(x_i)'}$

**Module** `Addressing`$(\mathcal{A})\,(n^{o(x_i)'}, n^{d(x_i)'})$:

  $r^{o(x_i)} \leftarrow f_r(o(x_i)'); r^{d(x_i)} \leftarrow f_r(d(x_i)')$

  $a^{o(x_i)} \leftarrow \beta \cdot (r^{o(x_i)})^T \mathcal{N}; a^{d(x_i)} \leftarrow \beta \cdot (r^{d(x_i)})^T \mathcal{N}$

  $a^{o(x_i)} \leftarrow SparseMax(a^{o(x_i)}); a^{o(d_i)} \leftarrow SparseMax(a^{o(d_i)})$

  **return** $a^{o(x_i)}, a^{d(x_i)}$

**Module** `Decoding`$(\mathcal{D})\,(\{a^{o(e_j),d(e_j)} \diamond \mathcal{M}\}, n^{o(e_j)} \oplus n^{d(e_j)}, W(S_G))$:

  $n^{e_j} \leftarrow n^{o(e_j)} \oplus n^{d(e_j)}$

  $c^{e_j} \leftarrow ContentBasedRead(\mathcal{M}, a^{o(e_j),d(e_j)})$

  $c^{e_j'} \leftarrow MinGain(c^{e_j}, n^{e_j})$

  $info \leftarrow Concatenate(c^{e_j}, c^{e_j'}, W(S_G))$

  $\hat{Q}_w(e_j) \leftarrow f_d(info)$

  **return** $\hat{Q}_w(e_j)$

**Function** `ChangeShape`$(Vector, \mathbb{R}^n, \mathbb{R}^m)$:

  Change shape of a vector from $\mathbb{R}^n$ to $\mathbb{R}^m$

  **return** vector

**Function** `ContentBasedRead`$(\mathcal{M}, a^{o(e_j),d(e_j)})$:

  $a^{o(e_j),d(e_j)} \leftarrow ChangeShape(a^{o(e_j),d(e_j)}, \mathbb{R}^{l_s \times l_s}, \mathbb{R}^{1 \times (l_s)^2})$

  $\mathcal{M}' \leftarrow ChangeShape(\mathcal{M}, \mathbb{R}^{l_s \times l_s \times 2l_n}, \mathbb{R}^{(l_s)^2 \times 2l_n})$

  $c^{e_j} = a^{o(e_j),d(e_j)} \mathcal{M}'$

  $c^{e_j} \leftarrow ChangeShape(m_j, \mathbb{R}^{1 \times 2l_n}, \mathbb{R}^{2l_n})$

  **return** $c^{e_j}$

**Function** `MinGain`$(c^{e_j}, n^{e_j})$:

  $n_1^{e_j} \leftarrow where(n^{e_j} > \epsilon, n^{e_j}, \epsilon)$

  $n_2^{e_j} \leftarrow where(n^{e_j} < \epsilon, MAX, 0)$

  $c^{e_j'} = [(c^{e_j} + n_2^{e_j})/n_1^{e_j}].min()$

  **return** $c^{e_j'}$

---

## A.2 Details of Training Algorithm for Mayfly

We give a detailed training algorithm of Mayfly in Algorithm 2.

---
**Algorithm 2:** Mayfly Training Framework

---
**Data:** Mayfly $(\mathcal{R}, \mathcal{A}, \mathcal{D})$ with all learnable parameters $\theta$, larval task set $\mathcal{T}^L$ or metamorphosis task set $\mathcal{T}^M$;

**for** *each task* $t_k : (s_k, q_k) \in \mathcal{T}^L/\mathcal{T}^M$ **do**

    Initialize $W(S_G^{(k)}) = 0$;

    **for** $x_i^{(k)} \in s_k$ **do**

        $\text{Store}(o(x_i^{(k)}), d(x_i^{(k)}), \Delta w_i^{(k)}, \mathcal{M})$;

        $W(S_G^{(k)}) += \Delta w_i^{(k)}$

    **for** $e_j^{(k)}, Q_w(e_j^{(k)}) \in q_k$ **do**

        $\hat{Q}_w(e_j^{(k)}) \leftarrow \text{Query}(o(e_j^{(k)}), d(e_j^{(k)}), \mathcal{M}, W(S_G^{(k)}))$;

        $L += \text{LossFunc}(Q_w(e_j^{(k)}), \hat{Q}_w(e_j^{(k)}))$;

    Backprop through queries and stores: $dL/d\theta$;

    Update learnable parameters: $\theta \leftarrow \text{Optimizer}(\theta, dL/d\theta)$;

    Normalize $\mathcal{N}$;

    Clear $\mathcal{M}$;

---

## A.3 Details of Algorithm for Larval/Metamorphosis Task Generation

The detailed generating algorithms for larval/metamorphosis meta-tasks are shown in Algorithm 3 and Algorithm 4, respectively.

---
**Algorithm 3:** Generating a Larval Task

---
**Data:** *Zipf* distribution pool $\mathcal{P}$; Stream length range $[1, \gamma]$;

**Result:** A larval task $t_k$;

Sample a stream length $|S_G^{(k)}|$ from $[1, \gamma]$;

Sample a total weight sum $W(S_G^{(k)})$ from $\left[5 \times |S_G^{(k)}|, 50 \times |S_G^{(k)}|\right]$;

Sample a distribution instance $p^{(k)} \sim P$;

**for** $i \in \left[1, |S_G^{(k)}|\right]$ **do**

    Systhesize a streaming edge $x_i^{(k)}$;

    Sample $p_i^{(k)} \sim p^{(k)}$ and $\Delta w_i^{(k)} \leftarrow W(S_G^{(k)}) \times p_i^{(k)}$;

    **add** $x_i^{(k)}$ to the $t_k$'s store set ($s_k$) with weight $\Delta w_i^{(k)}$;

Construct $t_k$'s query set ($q_k$) with all $(e_j^{(k)}, Q_w(e_j^{(k)}))$;

**return** a larval task $t_k$;

---

Table 4: Hyper-parameters Considered

| | |
|---|---|
| Learning rate | 1E-4,**5E-4**,1E-3 |
| Hidden size of $f_n$ | **32**,64 |
| Hidden size of $f_r$ | **32**,64 |
| Hidden size of $f_d$ | 32,**64**,128 |

## A.4 Visualization of Datasets

Table 6 shows the statistics of each datasets. We also present the distributional characteristics of Enron, LKML and Coauthor datasets on a log-log scale in Figure 8. These datasets exhibit varying

---

**Algorithm 4:** Generating a Metamorphosis Task

---

**Data:** Training set $D_{train}$;
**Result:** A metamorphosis meta-task $t_k$;

Sample a stream length $|S_G^{(k)}|$ from $[1, |D_{train}|]$;

Extract a continuous substream from $D_{train}$ with length $|S_G^{(k)}|$ as support set $s_k$;
Shuffle the correspondence between edges and their weights in support set $s_k$;
Multiply all weight $\Delta w_i$ in support set $s_k$ by a multiplier $y \in [1, 10]$;

Construct $t_k$'s query set $q_k$ with all $(e_j^{(k)}, Q_w(e_j^{(k)}))$;
**return** a metamorphosis task $t_k$;

---

Table 5: Settings of $\mathcal{N}$ and $\mathcal{M}$

| B | $l_s$ | $l_n$ | $l_r$ |
|---|---|---|---|
| 64KB | 32 | 8 | 16 |
| 128KB | 45 | 8 | 16 |

degrees of deviation from the ideal Zipf distribution, which is typically represented as a straight line in the figure.

## A.5 DETAILED SETTINGS OF EXPERIMENT

We did not deliberately tune the parameters of the Mayfly. Instead, we tried a few conventional neural network settings in Table 4 (best parameters are bolded) on an early instance of Mayfly and settled on one that strikes a balance between accuracy and efficiency. All the other experiments just simply followed the same setup. In addition, the default settings of $\mathcal{N} \in \mathbb{R}^{l_r \times l_s}$ and $\mathcal{M} \in \mathbb{R}^{l_s \times l_s \times 2l_n}$ under different budgets are shown in Table 5. Table 7 below presents a detailed breakdown of parameters for each network module, alongside the classic networks AlexNet and VGG16 for comparison. They are all small-scale neural networks with relatively low storage and training overheads. In addition, according to the consensus in neural data structures, the overheads of other modules can be amortized across different applications (Rae et al., 2019; Cao et al., 2023). After training, they can be copied to multiple application scenarios, analogous to the overhead of loading hash function libraries dynamically in TCM.

All of our experiments run at a NVIDIA DGX workstation with CPU Xeon-8358 (2.60GHz, 32 cores), and 4 NVIDIA A100 GPUs (6912 CUDA cores and 80GB GPU memory on each GPU).

## A.6 ADDRESSING MECHANISM

We gain insights into the addressing mechanism of the Mayfly by observing key variables in the addressing module during the larval phase. According to the properties of *Sparse SoftMax* (Laha et al., 2018; Martins & Astudillo, 2016), the 2-norm of $\beta$ continuously changes to control the sparsity of addresses (i.e., the proportion of non-zero bits in the vector), while the 2-norms of other variables, such as representation vectors and refined vectors, remain stable, as shown in Figure 9 (a). It is also evident that the 2-norm of $\beta$ increases progressively to encourage the sparsity of addresses, ultimately converging to 1, as shown in Figure 9 (a) and (b). It indicates that the Mayfly tends to allocate the cumulative information of an edge to only one slot of $\mathcal{M}$, thus alleviating information overlap. Moreover, Figure 9 (c) shows that the variance of the number of stored edges across multiple slots

| Datasets | Lkml | Enron | Coauthor | Twitter |
|---|---|---|---|---|
| Number of Nodes | 63,399 | 17,733 | 40,421 | 52,579,682 |
| Number of Streaming Edges | 1,096,440 | 475,097 | 2,668,965 | 10,929,205,002 |
| Graph Type | Communication | Communication | Co-authorship | Social |

Table 6: Statistics of Datasets

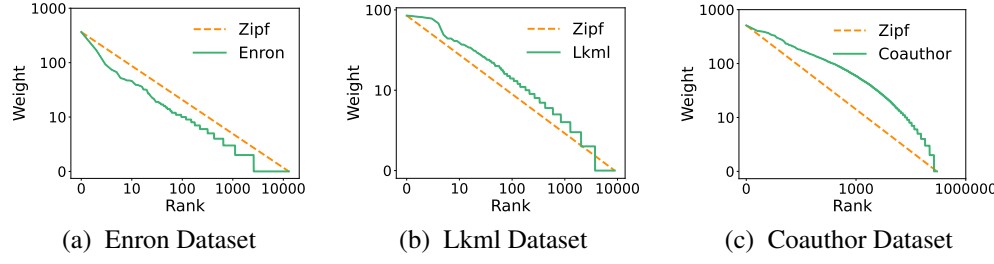

| (a) Enron Dataset | (b) Lkml Dataset | (c) Coauthor Dataset |

Figure 8: Visualization of Datasets

Table 7: Module Parameters

| Model Parameter | Mayfly64KB | Mayfly128KB | Mayfly256KB | AlexNet | VGG16 |
|---|---|---|---|---|---|
| Embedding Module | 1.93K | 1.93K | 1.93K | - | - |
| Addressing Module | 2.09K | 2.30K | 2.62K | - | - |
| Decoding Module | 5.27K | 5.27K | 5.27K | - | - |
| Total | 9.29K | 9.51K | 9.82K | 60907.38K | 1115786.64K |

in $\mathcal{M}$ continuously decreases, indicating its automatic learning of evenly storing edges to the set of slots to maximize the storage utilization.

### A.7 ABLATION STUDY

Ablation experiments, as shown in Figure 10, show that Mayfly with MinGain leads to faster convergence, enhanced stability, and superior training performance. We further conducted ablation experiments for Sparse softmax, which demonstrates that the Mayfly with Sparse softmax achieves faster convergence rates and better performance during training.

### A.8 THROUGHPUT

We evaluate the throughput of Mayfly in comparison with TCM and GSS in Table 8. Mayfly and TCM use a space budget of 64KB for processing Lkml of length 800,000. As an exception, GSS is allowed to allocate extra space if its 64KB budget is exhausted. For the implementation on CPUs, the Mayfly (Python) achieves a similar querying throughput to GSS (C++), while TCM (C++) performs the best, due to its simple hash-based structure. However, by employing simple parallel algebraic operations on GPUs, Mayfly achieves a remarkably high throughput, which is about 28 and 18 times higher than the storing and querying throughputs of TCM on CPUs. Mayfly may benefit from the growing trend of integrating GPU-based neural data structures in streaming applications, where inputs are either ephemeral or at high throughputs (Rae et al., 2019).

Table 8: Result of Throughput (Mops)

| Model | Store | Edge-query | Node-query | Model | Store | Edge-query | Node-query |
|---|---|---|---|---|---|---|---|
| Mayfly(GPU) | 64.01 | 54.98 | 27.09 | TCM(CPU) | 2.18 | 2.10 | 1.40 |
| Mayfly(CPU) | 0.09 | 0.43 | 0.13 | GSS(CPU) | 0.55 | 0.58 | 0.10 |

### A.9 ROBUST FOR DYNAMIC CONCEPT DRIFT

A simple approach for learned data structures leverages neural networks to predict and classify high/low-frequency items, thereby reducing hash conflicts and prediction errors (Hsu et al., 2019). However, this approach lacks robustness against dynamic drift in data streams. In contrast, our proposed Mayfly does not mechanically memorize the correspondence between specific items and their frequencies (see metamorphosis task generation in section 4.4), providing robustness against concept drift in dynamic stream scenarios.

To demonstrate the efficacy of the Mayfly on the dynamic drift, we implement an enhanced version of the TCM, termed LTCM, as per previous work (Hsu et al., 2019). The LTCM optimizes by capturing

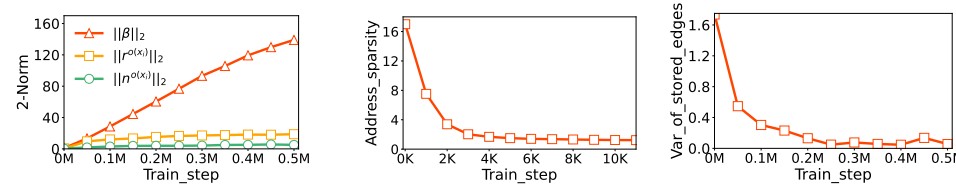

(a) $||\beta||_2$ ,$||r^{o(x_i)}||_2$ and $||n^{o(x_i)}||_2$ (b) Average Sparsity of Addresses (c) Average Variance of Edges

Figure 9: Addressing Analysis

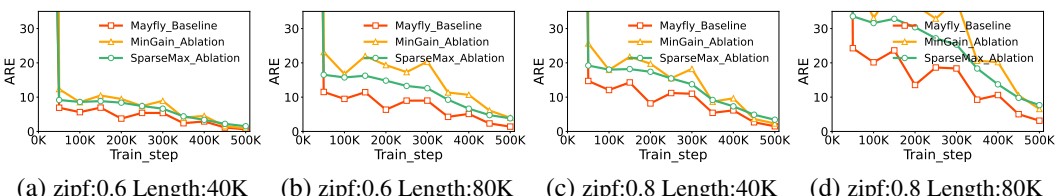

(a) zipf:0.6 Length:40K   (b) zipf:0.6 Length:80K   (c) zipf:0.8 Length:40K   (d) zipf:0.8 Length:80K

Figure 10: Ablation Study

1% and 5% of high-frequency terms. We select a real substream (B=64KB, Lkml) of length 80K from $D_{test}$, and progressively shuffle the item-frequency correspondence as shown in Figure 11. As the shuffle ratio increases from 0% to 100%, the AAE of the Mayfly fluctuates minimally between 2.40 and 2.62. In contrast, the AAE of LTCM (5%) starts at 8.34 and escalates to 10.60. This increase is attributed to the classifier of LTCM incurring more errors due to the dynamic shift, leading to a rise in estimation error and ultimately surpassing the original TCM's average AAE of 9.84.

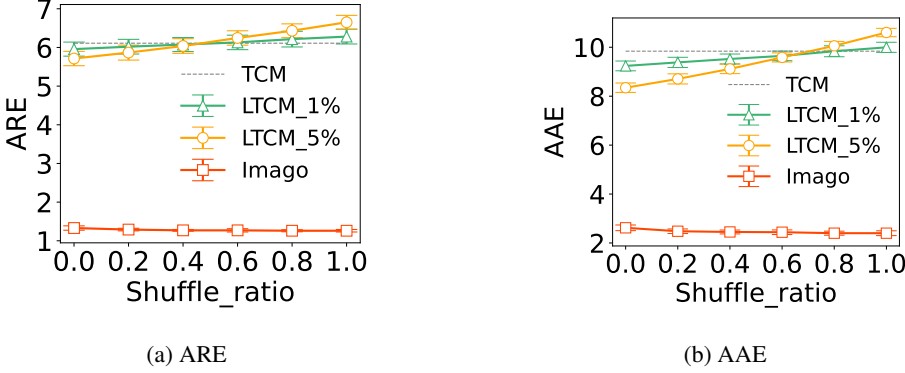

(a) ARE                    (b) AAE

Figure 11: LTCM vs. Mayfly

## A.10   DEFINITION OF COMPLEX QUERY

We provide formal definitions for path queries and subgraph queries (see Analysis Section).

**Path Query.** We adopt a prevalent path query, *maximum flow path query*, as detailed in Algorithm 5. The maximum flow in a path defined by a finite set of directed edges, corresponds to the minimum edge weight in the path. Note that the maximum flow defaults to 0 if any edge is not *exist*.

**Subgraph Query.** The Subgraph Query, as detailed in Algorithm 6, adheres to the query semantics of TCM. Its objective is to compute the sum of the weights of all edges in a subgraph $G_Q$ of the original graph $G$.

---

**Algorithm 5:** Path Query

---

**Data:** graph $G$; query path $\psi$; result = $MAX\_INT$
**Result:** the maximum flow in $\psi$
**for** $e \in \psi$ **do**
    **if** *not* $Q_c(o(e), d(e))$ **then**
        result=0;
        **break**;
    **if** $Q_w(e) < result$ **then**
        result = $Q_w(e)$;
**return** result;

---

**Algorithm 6:** Subgraph Query

---

**Data:** graph $G$; query subgraph $G_Q$; result = 0
**Result:** the aggregated weight of query subgraph $G_Q$
**for** $e \in G_Q$ **do**
    **if** *not* $Q_c(o(e), d(e))$ **then**
        result=0;
        **break**;
    **else**
        result += $Q_w(e)$;
**return** result;

---

### A.11 LIMITATION AND FUTURE WORK

As the inaugural neural data structure for graph stream summarization, the Mayfly presents substantial opportunities for enhancement. However, our focus is not on its intentional optimization. Insights into the workings of each Mayfly module can be found in the Analysis Section. For example, exploiting the prevalent sparsity in addresses could markedly improve storage and query efficiency. Moreover, integrating additional effective reading heads could notably boost query accuracy.

### A.12 SUPPLEMENTARY RESULTS

Due to the space constraint, the real-time performance of imago with $\alpha \in \{0.6, 0.8\}$ under concept shift is shown in Figure 12.

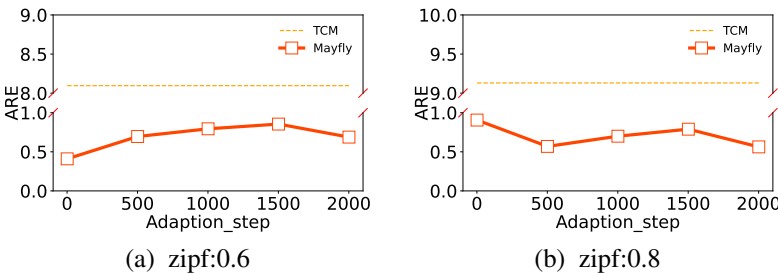

(a) zipf:0.6        (b) zipf:0.8

Figure 12: Other Concept Drifts

