# OpenReview forum: "Mayfly: a Neural Data Structure for Graph Stream Summarization"
_ICLR.cc/2024/Conference — ICLR 2024 spotlight_

### Official Review · Reviewer_AAng · 2023-10-31

**Soundness:** 4 excellent
**Presentation:** 3 good
**Contribution:** 4 excellent
**Rating:** 8
**Confidence:** 4

**Summary:**

The authors utilize memory-augmented networks for summarizing graph streams into a fixed space. Given the compressed result and a specific edge (or a subset of edges), one can approximate the edge's weight. The neural network is first pre-trained on synthetic graph streams and later fine-tuned using a fraction of the input stream. When compared to sketching-based approaches, the suggested method shows a better approximation accuracy for a given compression size.

**Strengths:**

S1. The overall design of the suggested proposed, particularly the use of memory-augmented networks for compressing graph streams, seems reasonable.

S2. The introduced method exhibits significant improvements compared to sketching-based approaches.

S3. The proposed successfully applies to billion-scale graphs, highlighting its practicality for real-world scenarios.

**Weaknesses:**

W1. The paper lacks a theoretical analysis of complexity and accuracy.

W2. The presentation could be clearer. The insights and novelties of the addressing, decoding, and store operations could be specified.

W3. The proposed method seems to only support weight-related queries, not graph algorithms like Dijkstra’s.

W4. While somewhat expected, the throughput of the proposed method is not as high as that of rule-based baseline methods.

**Questions:**

Q1. Can complex graph algorithms, like Dijkstra's, be executed using the summary? If they can, does it substantially impact the time complexity compared to executing the algorithms on the uncompressed graph?

Q2.  Can you provide a more detailed explanation of how the MiGain term contributes? The present explanation seems too concise.

Q3. What is the time complexity of the addressing, decoding, and store operations?

Q4. The reported ARE and AAE seem quite large. What are their values when we use the mean as the estimate of all edge weights?

---

> ### Author Response · Authors · 2023-11-16
> **Response to the Reviewer Part(1/2)**
>
> Thanks for your diligent responses and insightful suggestions. We have revised some statements in the manuscript based on your suggestions to make the paper clearer and more readable. We will address your concerns in this dialogue box and respond to your questions in the next one.
>
> ## W1(Q3). The paper lacks a theoretical analysis of complexity and accuracy.
> Thanks for your comments!
>
> In the training phase, Mayfly is trained on a series of auto-generated meta-tasks, so the time complexity of the model's training algorithm is mainly linearly related to the number and size of meta-tasks used in this phase. However, the number of meta-tasks does not affect the training space consumption as they are automatically generated, used, and released in a timely manner. Thus, the space consumed by training is primarily linear in the size of the meta-tasks.
>
> During the deployment phase, Mayfly aligns with conventional rule-based summarizations, exhibiting a constant time complexity O(1) for individual operations. Indeed, this is attributed to the fact that, upon the arrival of any edges, the neural network necessitates only a fixed amount of computation to ascertain the corresponding result. As for Mayfly's space consumption, it mainly depends on the memory size set according to the deployment requirements of applications.
>
> Regarding the theoretical guarantee of accuracy, as you correctly note, the neural data structures（i.e.,NBF, meta-sketch） grounded in memory networks lack a mature theory for analyzing their estimation accuracy. We agree that the theoretical guarantee is an important research direction for future research of neural data structures.
>
>
>
> ## W2. The presentation could be clearer. The insights and novelties of the addressing, decoding, and store operations could be specified.
> Thanks for your suggestions. We have revised Sections 1,4 and 6 of our paper to provide a more detailed emphasis on the novelty of the Mayfly. Below, we present the insights and novelties in the addressing, storage, and decoding aspects.
>
> In terms of addressing and storage, the Mayfly introduces a 2D separated in/out-degree addressing scheme, coupled with a novel joint compression storage for nodes and edges to consider the graph structural information, setting it apart from conventional neural data structures.
>
> Regarding decoding, Mayfly introduces a novel concept of altering information pathways in memory networks to enable multi-decoding for the expansion of query types (as detailed in Section 6).
>
> It eliminates the need for repetitive retraining of the neural data structure for each query type, simultaneously meeting the requirements of supporting multiple query types through one-pass storing in graph scenarios. This novel concept also holds the potential to facilitate research in other scenarios with diverse query types.
>
> For more innovations, please see the revised manuscript.
>
>
> ## W4. While somewhat expected, the throughput of the proposed method is not as high as that of rule-based baseline methods.
> We concur with your insightful perspectives. The additional operations in Mayfly and other neural data structures render them less competitive in efficiency. Meanwhile, we believe that the throughput of Mayfly can meet the needs of graph streaming scenarios. For instance, on CPUs, Mayfly operates at the same level of throughput (${10}^{-1}$ - ${10}^{0}$Mops) as graph sketches, such as GSS and TCM. Furthermore, Mayfly has good acceleration potential with GPUs due to parallel algebraic operations.

---

> ### Author Response · Authors · 2023-11-16
> **Response to the Reviewer Part(2/2)**
>
> ## Q1(W3). Can complex graph algorithms, like Dijkstra's, be executed using the summary?
> Yes, Dijkstra's algorithm can be executed using summarizations, Mayfly included, without effect the time complexity. However, we would be conservative in claiming that Dijkstra's algorithm can be fully solved by graph stream summarizations due to the precision of solutions.
>
> The key idea behind Dijkstra's algorithm is to iteratively select the vertices with the currently shortest known distance from the start node and update the distances to its neighboring nodes. Thus it is feasible to run Dijkstra’s algorithm on top of graph stream summarizations instead of the original graphs through iterative invocations of two distinct types of atomic queries: edge weight and connectivity queries. In terms of complexity, it is known that Dijkstra's algorithm has a time complexity of O((V + E) log(V)), if the graph’s adjacency list/matrix can be accommodated in the main memory. So, following the procedure of Dijkstra's algorithm, the complexity will not be reduced. Neither will it be increased, as the complexity of atomic operation in graph stream summarizations is O(1).
>
> However, we cannot assert that current graph stream summarizations in restricted space budgets, e.g. Mayfly and TCM, could fully solve complex graph algorithms such as Dijkstra's.  These iterative invocations of atomic queries to support complex algorithms by graph stream summarizations lead to the accumulation of estimation errors. These cumulative errors ultimately lead to a progressive deviation from the ground truth. This problem may be alleviated if future research builds on Mayfly to tailor direct solutions directly to a single complex algorithm, e.g., in combination with changing the information pathway to add a customized decoding module. In contrast, conventional graph stream summarizations do not have this potential because they are designed based on human-defined rules.
>
> ## Q2. Can you provide a more detailed explanation of how the MiGain term contributes?
>
> We apologize for any confusion and have clarified the statement in the revised manuscript. The MinGain term utilizes information extracted from memory while concurrently leveraging edge embedding information for information extraction and denoising. The fundamental idea behind this is to extract low numerical value bits of information crucial for decoding but easily overshadowed by stacked noise from other large numerical value bits. The detailed formalization of the MinGain is articulated in Algorithm 1 of Appendix A.1. Furthermore, the ablation experiments are provided in Appendix A.7
>
>
> ## Q4. The reported ARE and AAE seem quite large. What are their values when we use the mean as the estimate of all edge weights?
> Thanks for your comments. Our experiments cover a range of space constraints, from relatively abundant to very tight, providing a nuanced perspective of Mayfly. Thus, in several very tight budget situations, all methods show large ARE and AAE, while Mayfly's significantly lower errors in comparison demonstrate excellent robustness.
>
> Furthermore, employing mean estimation is not feasible in graph streaming scenarios. Primarily, this approach can’t support other graph query types in Section 6 of our paper due to a significant loss of graph structures. Moreover, in settings where edges in streams are reached repeatedly, calculating the mean of weights is challenging, given that determining the number of distinct edges in the streams incurs a considerable cost. The table below presents the performance comparison of Mean Estimator and Mayfly. Even ignoring the above factors, the errors of the mean predictor are still larger than those of Mayfly across all distributions with varying skewness.
>
>
> |ARE Metric|$\alpha$=0.2|$\alpha$=0.6|$\alpha$=1|
> |-|-:|-:|-:|
> |Mean Estimator|1.01|1.8|5.71|
> |Mayfly 64KB|0.44|0.97|4.82|
> |Mayfly 128KB|0.39|0.38|2.28|
>
> |AAE Metric|$\alpha$=0.2|$\alpha$=0.6|$\alpha$=1|
> |-:|-:|-:|-:|
> |Mean Estimator|5.62|7.7|13.58|
> |Mayfly 64KB|2.54|4.68|10.41|
> |Mayfly 128KB|2.4|2.83|5.55|

---

> > ### Comment · Reviewer_AAng · 2023-11-21
> > **Thank you**
> >
> > Thank you for the clarifications. I would like to maintain my positive score.

---

### Official Review · Reviewer_GvCp · 2023-11-01

**Soundness:** 4 excellent
**Presentation:** 4 excellent
**Contribution:** 4 excellent
**Rating:** 8
**Confidence:** 3

**Summary:**

This work proposes a neural data structure for graph stream summarization, which has neatly designed modules (representation, addressing and decoding), as well as corresponding operations such as store and multiple types of queries. Experiments on Lkml and Enron datasets demonstrate the superiority of the designed approach compared to the SOTA TCM and GSS baselines.

**Strengths:**

1. The paper studies an interesting topic on using neural data structure for graph stream summarization, and designed the approach very neatly. In general, the presentation of the methodology is very clear and makes it easier to understand the nontrivial details.

2. The training strategy is inspiring, following the paradigm of "pre-training and fine-tuning", and thus has two stages called larval phase and metamorphosis phase.

3. Comprehensive experimental results demonstrate both the effectiveness of the method on different types of queries and throughputs.

**Weaknesses:**

1. Minor: A table of dataset statistics is suggested to make it more straightforward.

**Questions:**

1. In addition to throughput, is there any other metric to evaluate the efficiency of the method (e.g., query time)?
2. How to balance between the parameter size and model accuracy?

---

> ### Author Response · Authors · 2023-11-16
> **Response to Reviewer GvCp**
>
> Thanks for your valuable suggestions and constructive reviews. We've added a table showing the statistics of the datasets to make it more straightforward. Here are our responses to your questions:
>
> ## W1. Minor: A table of dataset statistics is suggested to make it more straightforward.
> Thanks for your suggestions! We have added a table of the dataset statistics as follows and put it in Appendix A.4. In addition, we present the distributional characteristics of datasets showcasing the fact that the datasets exhibit varying degrees of deviation from the ideal Zipf distribution, in Appendix A.4.
>
> | Datasets | Lkml | Enron | Coauthor | Twitter |
> |---|---:|---:|---:|---:|
> | Number of Nodes | 63,399 | 17,733 | 40,421 | 52,579,682 |
> | Number of Streaming Edges | 1,096,440 | 475,097 | 2,668,965 | 10,929,205,002 |
> | Graph Type | Communication | Communication | Co-authorship | Social |
>
> ## Q1. In addition to throughput, is there any other metric to evaluate the efficiency of the method (e.g., query time)?
>
> Thanks for the insightful question! Yes, we agree that store/query time, referred to as latency, is another aspect of evaluating efficiency. However, consistent with other graph sketches, Mayfly focuses more on the estimation accuracy and processing throughput.
>
> In the table below, we compare the latency of Mayfly and baselines TCM and GSS, under the same settings of **throughput experiments**. Despite not surpassing conventional rule-based methods, Mayfly effectively maintains the latency of a single operation consistently around 1 ms, which is sufficient for existing graph streaming applications.
>
>
> | singular Latency(ms) | Store | Edge-query | Node-query |
> |---|---:|---:|---:|
> | TCM | 0.00200 | 0.00199 | 0.00324 |
> | GSS | 0.01743 | 0.01854 | 0.03998 |
> | Mayfly | 0.81083 | 0.78387 | 0.90593 |
>
> In addition, serial writes/queries of individual edges are often uneconomical in practice. Hence, a small buffer is usually established, and edges are stored into sketches once it fills. This setup allows Mayfly to leverage concurrent writes/queries, resulting in a near-linear reduction in average latency, as depicted in the table below. It shows that as the buffer size expands, Mayfly maintains a linear reduction trend in average latency, while baselines see marginal benefits.
>
> | Average Store Latency  w.r.t. Buffer Size  | 1 | 4 | 16 | 64 |
> |---:|---:|---:|---:|---:|
> | TCM | 0.00200 | 0.00134 | 0.00117 | 0.00103 |
> | GSS | 0.01743 | 0.01352 | 0.01152 | 0.01081 |
> | Mayfly | 0.81083 | 0.17610 | 0.05180 | 0.02604 |
>
> | Average Edge-query Latency w.r.t. Buffer size  | 1 | 4 | 16 | 64 |
> |---:|---:|---:|---:|---:|
> | TCM | 0.00199 | 0.00142 | 0.00118 | 0.00108 |
> | GSS | 0.01854 | 0.01524 | 0.01210 | 0.01138 |
> | Mayfly | 0.78387 | 0.21451 | 0.19491 | 0.07470 |
>
> ## Q2. How to balance between the parameter size and model accuracy?
>
> Empirical studies show that increasing the parameter size of the neural module within a certain range correlates with enhanced model accuracy. However, once the module parameter sizes reach specific thresholds, the improvement in model accuracy becomes exceedingly marginal. Considering the running overhead, we chose a set of parameters below these thresholds to balance accuracy and efficiency.
>
> In Appendix A.5, Table 6, we present the current parameter size for each module. It shows that the parameter sizes of their respective modules in Mayfly are consistently kept on a small scale. Also, with the expansion of Mayfly's memory size, the parameter size of every module remains stable. The addressing module is the exception, requiring a slight increase in parameter size to align with the number of memory slots in the memory module.

---

### Official Review · Reviewer_wDZJ · 2023-11-01

**Soundness:** 2 fair
**Presentation:** 3 good
**Contribution:** 2 fair
**Rating:** 6
**Confidence:** 4

**Summary:**

The authors propose a Mayfly framework consisting of two stages (larval and metamorphosis) for summarizing graph streams. It is the first neural data structure for graph stream summarization. The Mayfly acquires basic summarization capabilities by learning from synthetic data and can be rapidly adapted to real graph streams. The Mayfly framework is agile and customizable, supporting a broad range of graph queries with lightweight information pathway configurations.

**Strengths:**

S1: The authors' integration of Mayfly with machine learning is interesting.
S2: Well-written.

**Weaknesses:**

W1: It seems that this paper leans more towards an engineering-oriented research, and its technology appears to be quite fundamental. For instance, it utilizes techniques like meta-learning and follows the paradigm of pre-training and fine-tuning. In my view, there's very little innovation at the model level, which is the most fundamental reason for my minor rejection.
W2: The authors' integration of graph streams with the Mayfly framework is indeed intriguing. However, from my perspective, it appears to be a fundamental application of pre-training and fine-tuning methods. I believe the authors should provide a clear explanation in the introduction or methodology section regarding why the Mayfly framework is particularly suitable for adapting to graph data streams.
W3: While the authors propose the Mayfly framework with the aim of enhancing data stream processing efficiency, I did not see a clear experimental demonstration of its efficiency gains. In other words, I would appreciate a more tangible comparison between the Mayfly framework and the comparative algorithms in terms of actual runtime efficiency under the same spatial budget.

**Questions:**

Please refer to weaknesses.

---

> ### Author Response · Authors · 2023-11-16
> **Response to Reviewer wDZJ**
>
> Thanks for your reviews. Your feedback is important to us. We've revised our manuscript based on your suggestions to more clearly articulate our novelties.
>
> ## W1&W2: "Little innovation at the model level" & "Why the Mayfly framework is particularly suitable for adapting to graph data streams?"
>
> Thank you for your comments. We have revised Sections 1,4 and 6 to highlight the novelty of the Mayfly in adapting to graph streams. We will uniformly respond and address your two concerns as below.
>
> We argue that Mayfly, as the first neural graph sketch, innovates on multiple levels (including at the model level), aiming primarily at addressing graph stream scenarios. Graph sketches present greater challenges compared to conventional sketches. For example, conventional sketches count only entity frequencies, whereas graph sketches also tally interaction frequencies and enable diversity queries. It thus asks for novel techniques to address the challenges in different aspects, such as network structures, training strategies, and versatility in supporting various graph queries.
>
>
> **Novelty in Network Structures.** The Mayfly follows the general framework of memory-augmented networks but is uniquely tailored for graph stream scenarios by introducing a 2D separated in/out-bidirectional addressing scheme, coupled with a novel method of storing edges/nodes information within the memory module. Mayfly takes into account the graph structural information and employs joint compression storage for nodes and edge weights, which is unseen in neural data structures.
>
> **Novelty in Training Strategies.** Similar to existing neural data structures, such as Meta-sketch, Mayfly employs a 'pre-training and fine-tuning' paradigm. Nevertheless, Mayfly stands out by conducting pre-training totally independent of real data, utilizing randomized numeric IDs for nodes and Zipf distribution for weights. Meanwhile, the Training Strategy of Mayfly is designed for graph streams and supports a diversified set of queries (e.g., node, connectivity, path, and subgraph queries). We proposed automated meta-task generation algorithms for different atomic graph queries, providing inspiration for the development of neural data structures in the field of graph stream processing.
>
> **Novelty in Multi-typed Query Processing Paradigm.** The Mayfly introduces a novel concept of altering information pathways in memory networks to facilitate the extension to different query types (as detailed in Section 6). It eliminates the need for repetitive retraining of the neural data structure for each query type, simultaneously meeting the requirement of supporting multiple query types through one-pass storing in graph scenarios. This novel concept also holds the potential to facilitate research in other scenarios with diverse query types. In addition, it also provides certain interpretability to the execution mechanism of neural data structures.
>
> Notably, we have not deliberately optimized certain modules or training algorithms within Mayfly but have employed simple, general methods to demonstrate its effectiveness as a foundational paradigm.
> Meanwhile, we explore the addressing and memory mechanisms of Mayfly in Appendix A.6 and A.9, providing comprehensive and insightful guidance for further model optimization. Moreover, complex designs at the model level may not yield performance gains due to the trade-off with corresponding storage overhead. Therefore, based on empirical studies, we have chosen the current simple yet effective solution for Mayfly.
>
>
> ## W3: While the authors propose the Mayfly framework with the aim of enhancing data stream processing efficiency...
>
> First, we clarify that Mayfly, like existing neural data structures, does not focus on enhancing efficiency. Instead, neural data structures are dedicated to capturing patterns of stream distributions to improve estimation accuracy.
>
> Second, Mayfly fully meets the efficiency requirements of graph stream applications. In terms of execution complexity, whenever a streaming edge arrives, Mayfly only needs to call its network modules to perform a finite number of computations to complete an individual operation, and thus it provides a constant time complexity O(1), consistent with conventional sketches. Additionally, Mayfly's network modules utilize shallow networks with small-scale parameters to further ensure computational efficiency, as demonstrated in Appendix A.5. In terms of store/query throughput, detailed experiments are presented in Appendix A.8. The results show that: on CPUs,  Mayfly operates at the same level of throughput ($10^{-1}$-$10^0$ Mops) as graph sketches, such as GSS and TCM; On GPUs, Mayfly achieves higher throughput by employing simple parallel algebraic operations.

---

> > ### Comment · Reviewer_wDZJ · 2023-11-22
> > **discussion**
> >
> > i appreciate the authors' response and will raise my score to 6 while the authors are suggested to further improve this work accordingly in the final version.

---

### Meta-Review · Area_Chair_z44e · 2023-12-07

**Metareview:**

This paper proposes a new Neural Network-based data structure, Mayfly, for graph summarization problem where edge data arrive in a streaming manner. With its two meta-learning based offline training stages, Mayfly, achieves better accuracy and adaptivity for many different types of graph queries on billion-scale graphs. Authors are encouraged to incorporate the reviewer comments and improve clarity in the next revision.

**Justification For Why Not Higher Score:**

Clarity can be improved according to reviews.

**Justification For Why Not Lower Score:**

Practical work on an interesting data domain.

---

### Decision · Program_Chairs · 2024-01-16

Accept (spotlight)